# The Clinical Utility of Assessing Orthostatic Intolerance in Adolescents with Sport-Related Concussion, a Retrospective Study

**DOI:** 10.3390/diagnostics14232743

**Published:** 2024-12-05

**Authors:** Mohammad N. Haider, Jule Deren, Karim Khaled, Cathlyn Corrado, Haley M. Chizuk, Jeffrey C. Miecznikowski, John J. Leddy

**Affiliations:** 1UBMD Orthopaedics and Sports Medicine, Jacobs School of Medicine and Biomedical Sciences, State University of New York at Buffalo, Buffalo, NY 14221, USA; 2Department of Neurology, Jacobs School of Medicine and Biomedical Sciences, State University of New York at Buffalo, Buffalo, NY 14068, USA; 3Department of Chemistry, John Carroll University, University Heights, OH 44118, USA; 4Department of Pediatrics, University of Rochester Medical Center, Rochester, NY 14642, USA; 5Department of Biostatistics, School of Public Health and Health Professions, State University of New York at Buffalo, Buffalo, NY 14068, USA

**Keywords:** sport related concussion, adolescent, exercise intolerance, orthostatic intolerance, orthostatic hypotension

## Abstract

Background/Objective: Assessing Orthostatic Intolerance (OI, symptoms upon standing from supine) is recommended in athletes with sport-related concussions (SRCs), as this is caused by impairments in the cardiovascular autonomic nervous system (cANS). Early Exercise Intolerance (Early EI, symptoms on light physical exertion) is also due to impairments in the cANS but is difficult to incorporate into outpatient clinical practice (cost of personnel, time, equipment). The purpose of this study was to determine if we could use OI to screen for Early EI, as well as understand differences between adolescents who do and do not report OI. Methods: Retrospective chart review. Adolescents with physician-diagnosed SRC performed the 2 min supine to 1 min standing OI test and the Buffalo Concussion Treadmill Test (BCTT) during their first post-SRC visit. Early EI was defined as more-than-mild symptom exacerbation at a heart rate (HR) below 135 bpm on the BCTT; OI was defined as new or increased symptoms of dizziness or lightheadedness on postural change. The sensitivity, specificity and diagnostic accuracy were calculated. Participants with and without OI were compared. Results: In total, 166 adolescents (mean 15.4 years, 58.8% male) were seen a mean of 5.5 days after injury; 48.2% had OI and 52.4% had Early EI, but there was no association between the two measures (Phi = 0.122, *p* = 0.115). The sensitivity and specificity (with 95% confidence intervals) for OI to screen for Early EI were 54.0% (43.5, 64.3) and 58.2% (47.2, 68.7), respectively. Adolescents with OI had a higher incidence of delayed recovery (24% vs. 9%, *p* = 0.012). Conclusions: Although both measures seem to be related to impaired autonomic function after SRC, OI has limited accuracy in screening for Early EI, which suggests that their etiologies may be different. Nevertheless, the assessment of OI has clinical utility in the management of SRC.

## 1. Introduction

Sport-related concussion (SRC), a form of mild traumatic brain injury, is challenging to diagnose due to its non-specific symptomatology and the lack of imaging findings or blood biomarkers specific to the disease [1]. It is therefore recommended that clinicians assess several domains that can be affected by concussion [2]. One of these assessments is a screening for Orthostatic Intolerance (OI), which presents as increased symptoms of dizziness or lightheadedness on postural change [3,4] and is a predictor of delayed recovery [5]. The primary cause of OI is impairment of the cardiovascular Autonomic Nervous System (ANS), and reduced baroreceptor sensitivity and attenuated blood pressure (BP) control have been identified in patients with acute concussion [6,7]. However, postural change can also illicit other concussion-like symptoms, which are discussed further in the manuscript. Post-concussion OI can be assessed easily in the clinic using a 2 min supine to 1 min standing OI screening [8], which is part of the Sport Concussion Office Assessment Tool (SCOAT) developed by the 2022 Amsterdam Consensus Conference Statement on Concussion in Sport [2].

Another recommended, but not mandatory, component is an assessment of the patient’s ability to exercise. Exercise tolerance is commonly assessed using graded exertion testing [9]. Concussion-related exercise intolerance, defined as a more-than-mild worsening of symptoms such as headaches, dizziness or excessive fatigue during exertion, is a characteristic finding in adolescent athletes after SRC [9] and is also suspected to be due to impairments in the cardiovascular ANS. However, there are other causes of exercise-induced symptoms shown during exertion testing, and these are discussed later in the manuscript. A systematic review and meta-analysis showed that approximately 95% [10] of athletes exhibit some degree of exercise intolerance on exertion testing within 2 weeks of SRC. The degree of exercise intolerance early after SRC is also prognostic of recovery duration [11]. One study [12] found that Early Exercise Intolerance (Early EI), defined as experiencing more-than-mild symptom exacerbation below a heart rate (HR) threshold (HRt) of 135 bpm on the Buffalo Concussion Treadmill Test (BCTT), was associated with a significant risk of delayed recovery. Two randomized controlled trials (RCTs) [13,14] of prescribed targeted heart rate aerobic exercise based on the results of the BCTT reported faster recovery from SRC, and one reported a significant reduction in the incidence of delayed recovery (i.e., of Persisting Post-Concussive Symptoms [1], PPCS). It has also been reported that those with the greatest degree of exercise intolerance early after SRC benefit the most from individualized aerobic exercise treatment [15]. While systematic exertion testing is useful for the diagnosis, treatment, and determination of physiological recovery from SRC, it is not without limitations. Exertion testing requires dedicated space, a trained clinician (e.g., athletic trainer, physical therapist, exercise physiologist, or equivalent), equipment (a treadmill or cycle ergometer), and the patient’s time (30 min or more) [16].

It would improve clinical efficiency if clinicians had a way to conveniently identify which patients would likely have Early EI after SRC and would benefit most from performing graded exertion testing and from receiving individualized aerobic exercise treatment. We have shown that 37% of adolescents demonstrate OI [17] while 30–52% have Early EI within 10 days of SRC [11,12], and that both impairments are predictors of delayed recovery [5,11]. Since both Early EI and OI are suspected to reflect cardiovascular autonomic impairment, it may be possible to use the convenient OI test to screen for Early EI. Hence, the purpose of this study was to see if we could use the OI test to screen for Early EI, as well as explore differences between adolescents with and without OI. We hypothesized that the OI test would have excellent (>90%) sensitivity and specificity in screening for Early EI.

## 2. Methods

The current study is a reanalysis of data extracted from our practice’s retrospective concussion patient registry [5] and excludes participants whose orthostatic data have been published previously [17]. Data were extracted from one university-affiliated sports medical practice that has a dedicated concussion management clinic. Participants were seen by physicians who diagnosed SRC using a relevant history, a concussion-focused physical examination, and any adjunct tests as needed (i.e., a graded exertion test) [2]. High school student athletes (i.e., those requiring clearance for return-to-play scholastic sport) and athletic adolescents with recreation-related SRC performed the BCTT at their initial visit as part of our standardized clinical management protocol. Participants were seen weekly by their treating physician for the first four weeks and then once or twice a month as needed until clinical recovery. The dates of injury and recovery were obtained from clinic notes. The definition of recovery was standardized among all physicians: the resolution of concussion-like symptoms to baseline, a physical examination within normal limits, and the ability to exercise to ≥80% of the age-appropriate HR maximum without the exacerbation of concussion-like symptoms [18].

### 2.1. Participants

The inclusion criteria were as follows: (1) adolescents (aged 13–18 years) (2) presenting within 10 days of injury and (3) diagnosed with SRC (ICD 10 code: S06.0X0A and S06.0X1), according to international criteria [1]. Patients at our practice were not diagnosed with an SRC if their current injury was more severe than mild, as indicated by a score < 13 on the Glasgow Coma Scale, lesion on CT/MRI, a focal neurologic sign consistent with an intracerebral lesion (S06.1-6XXA), and/or injury involving the loss of consciousness for 30+ min or post-traumatic amnesia for 24+ h (S06.0X3-6A). Participants were not included in the analyses if they had missing orthostatic or exertion testing information.

### 2.2. Main Outcome Measure

An automated cuff (Welch Allyn Connex ProBP 3400, Skaneateles Falls, NY, USA) was used to measure the participants’ HR and BP during a 2 min supine to 1 min standing OI test [17]. The first measurement was taken after the patient was resting supine for 2 min and the second measurement was taken after the patient was standing for 1 min. Physicians recorded if the patient reported lightheadedness (a sensation of fainting) or dizziness (a motion sensation) while supine and immediately and one minute after standing. HR and BP were measured again after one minute of standing. Changes in the participants’ systolic and diastolic BP and HR were calculated by subtracting the supine values from the standing values. Exercise tolerance was measured using the BCTT [11]. Before beginning the BCTT, participants rated their symptoms on a Visual Analogue Scale (VAS, 0–10) [9] to determine the initial symptom severity, and their resting HR was measured in a seated position after a 2 min rest using a Polar HR monitor (OH1 and Verity Sense, Kempele, Finland). The participant then walked on a level treadmill (Landice L8, Randolph, NJ, USA) at 3.2 mph (3.6 mph in participants 5′10″ and above) at 0° incline. The incline was increased by 1° each minute for the first 15 min unless the participant was unable to continue due to symptoms or excessive fatigue [9] The participants’ HR at exercise cessation was recorded as the HRt.

Early Exercise Intolerance: Early EI was defined as having a more-than-mild (≥3 points on a 0–10 pain visual analog scale) exacerbation of concussion symptoms at a HR of ≤ 135 bpm [12].

Orthostatic Intolerance: OI was defined by the patients’ report of new onset or increased symptoms of lightheadedness or dizziness upon standing, regardless of changes in vital signs [17]. Additionally, Orthostatic Hypotension (OH) was defined as a 20 mmHg or greater reduction in systolic BP, or a 10 mmHg or greater reduction in diastolic BP after 1 min of standing from supine [7].

### 2.3. Additional Clinical Measures

Self-reported nausea, balance problems, and dizziness rated on a 0–6 Likert scale were obtained from concussion symptom checklists (Post Concussion Symptom Inventory, PCSI) [19]. The results of the Vestibular Ocular Reflex (VOR) and Complex Tandem Gait (CTG) tests were obtained from the physician’s notes [20]. For VOR, any abnormal saccadic eye movements, a very slow performance, an inability to maintain visual fixation (i.e., beating back to the center), or the provocation of dizziness or headache were considered abnormal [20]. For CTG, an inability to walk in a straight line, extreme truncal sway, stumbling, or stepping out of line with eyes open were considered abnormal, but one misstep or minimal sway during the eyes-closed portion of the test was considered within normal limits.

### 2.4. Statistical Analysis

Univariate statistics were used to describe the sample. Sensitivity, specificity, the positive predictive value (PPV) and the negative predictive value (NPV) were estimated, along with 95% confidence intervals (CIs) [21]. The Yule’s Phi Coefficient was calculated to assess the association between OI and Early EI, and interpreted using guidance from Njuka et al. [22]. The sample was then stratified by the presence of OI and the clinical characteristics were compared. Continuous variables were compared using independent two sample *t*-tests and categorical variables were compared using a chi-squared test. A *p*-value of <0.05 was considered statistically significant. All analyses were performed using SPSS Version 29 (IBM Corp., Armonk, NY, USA).

## 3. Results

From January 2018 to December 2019, 952 patients with concussion were seen; out of these, 166 adolescents are included in the analysis. Figure 1 provides the sample inclusion flowchart.

The demographics and clinical characteristics are provided in Table 1. Participants were seen approximately 5.5 days after injury, included a slightly higher proportion of males, and 16% had symptoms that lasted for more than 4 weeks (i.e., developed PPCS).

Table 2 provides a 2 × 2 table presenting the results of OI and Early EI. The sensitivity of OI for Early EI was 54.0% (43.5, 64.3) and the specificity was 58.2% (47.2, 68.7). The PPV was 58.8% (47.8, 69.1) and the NPV was 53.5% (43.0, 63.8). OI and Early EI did not have a significant association, and the strength of this coefficient was negligible (Phi = 0.122, no or negligible range = 0.00–0.19) [22].

Table 3 presents the comparison of adolescents with and without OI. Adolescents with OI did not differ in their demographics, vital signs or exertion testing results, but took longer to recover and had a higher incidence of delayed recovery. Adolescents with OI also reported greater symptoms and had more abnormal VOR tests.

## 4. Discussion

Easy, cost-effective and accurate screening tests are useful for clinical practice; hence, we examined the diagnostic accuracy of using a brief OI test to screen for Early EI on graded exertion testing in adolescents within 10 days of SRC. Although roughly half of our sample had OI and half had Early EI, there was no association between the two measures. Our PPV was 58.8%, which means that the clinician would need to test 10 patients to identify ~6 abnormal results. This PPV may be an acceptable screening tool for cancers [23], for example, but not for predicting Early EI after SRC; this is because almost all adolescents would need to be tested to find the half that will likely have Early EI. In our sample, adolescents with and without OI did not differ in their pre-injury demographics, exertion testing results or HR/BP assessments, which is similar to our prior report [17]. Adolescents with OI, however, reported more balance-related symptoms, had more VOR abnormalities, and took significantly longer to recover. Since symptoms persisting beyond 28 days after concussion significantly adversely affect adolescents’ quality of life and school performance [24,25], it is important for clinicians at the initial visit to try to identify those at a greater risk of delayed recovery so that they may implement evidence-based treatment recommendations for SRC as soon as possible (e.g., avoiding strict rest, early light physical activity, prescribed aerobic exercise, sleep hygiene) [26]. The OI test can identify those at risk of delayed recovery and is easy to integrate into a comprehensive clinical evaluation; therefore, our study supports that it should be part of the SCOAT for the best-practice management of SRC [2].

The cardiovascular ANS regulates HR and BP in response to orthostatic and exercise challenges [27]. Orthostatic and exercise intolerance may, however, reflect different aspects of ANS impairment after SRC. During progressive exercise, HR and BP are mediated by both sympathetic (SNS) and parasympathetic (PNS) nervous system activation, albeit in opposing directions [28]. At lower intensities, the PNS dominates; however, at higher intensities, PNS withdrawal occurs and the SNS takes over. The exact etiology of EI in adolescent athletes early after SRC is not known. It has been suggested that aerobic deconditioning due to restriction from sport [1] produces EI, but this does not explain its occurrence within 10 days of injury since athletes do not show physiological signs of deconditioning until at least 2 to 3 weeks after they have stopped training [29]. The more likely explanation is the improper activation of the cardiovascular ANS to meet increased cardiovascular demands from the periphery during exercise, and several studies have shown abnormal HR and heart rate variability (HRV, a marker for ANS tone) responses during exercise early after SRC [30,31]. Autonomic dysfunction appears to impair cerebral blood flow regulation as well, which also may contribute to concussion-related EI [32,33].

Unlike Early EI, the etiology of physiological OI is well understood [3]. OI is due to the hypoperfusion of the brain during postural change, leading to symptoms faintness, lightheadedness or dizziness [7]. Baroreceptor sensitivity reflects the capacity of the cardiovascular ANS to accommodate dynamic metabolic demands in the periphery, and experimental studies have identified alterations in BP variability [34] and the attenuation of baroreceptor sensitivity [6] in concussed athletes within 5 days of injury. OI is primarily due to delayed SNS activation, not PNS withdrawal [35,36], which may help explain the lack of association between OI and Early EI. Additional research is warranted to understand why OI is not associated with Early EI.

Another reason for the lack of agreement between these two autonomic measures is that there are other, non-cardiovascular causes for symptom exacerbation in these tests, and OI and EI can be presented in non-concussed adolescents. Concussed adolescents who report OI may experience dizziness due to post-concussion vestibular dysfunction that is exacerbated on postural change, rather than hypoperfusion of the brain [37]. This is supported by our results, since adolescents with OI also had more abnormal balance findings. Similarly, symptom worsening during treadmill testing can also be due to balance problems [9]. Other causes of symptoms during OI or EI testing include cervical strains and migraine-type headaches that worsen with motion and physical activity.

### Alternate Screening Tests for Exercise Intolerance

It could be argued that the best way to screen for Early EI would be to exercise test only those patients who report that physical activity or exercise exacerbate their concussion symptoms. However, this may not be accurate for several reasons. Athletes with concussion are told not to participate in sport, so they may not have tried to exert themselves and do not know if their symptoms get worse with moderate or higher levels of exertion, leading to false negatives. On the other hand, patients may report exercise intolerance if they cannot exercise for other reasons, such as musculoskeletal injury or a self-perceived inability to exercise, leading to false positives. Since Early EI is suspected to be due to ANS dysfunction, additional screening tests for ANS function could be relevant. Resting HRV over a 5 min recording is a reliable marker for ANS that can be measured conveniently using commercially available HR monitors [38]. An abnormal HRV has been reported in some studies of SRC [39,40,41,42], yet others have found no association or report contradictory findings [6,31,34,43,44,45]. Resting HRV is highly variable and is confounded by several factors, including emotional state, alertness/fatigue, time since last food or drink, medications, and circadian rhythm [46]. Controlling these factors in a clinical setting is difficult. Another option is assessing the HRV response to known stimuli, such as face cooling [43] or the cold pressor test [47]. These tests reduce variability but require sophisticated equipment. The same caveats apply to other validated tests of ANS function, such as tilt testing, microneurography, and sweat testing [48,49].

## 5. Limitations

As a retrospective chart review, we were able only to identify associations between these clinical measures and not causation. Forty-two adolescents did not perform the BCTT at the initial assessment, which may have affected our results. The reason for not performing the BCTT was not recorded, but common reasons include patient refusal due to improper attire or a lack of time, and a lack of available staff to perform the BCTT. Additionally, since this study was performed in an out-patient clinical setting, we did not control for a variety of factors that can affect HR and BP, such as current pain, emotional state, time since food or drink intake, and sleep/fatigue. We assumed that these would randomly affect our sample, but future prospective studies should attempt to control for them. Our sample of athletic adolescents within 10 days of injury may not generalize well to older or younger patients, those experiencing persisting symptoms, or the non-athletic population. Lastly, there is variability in cardiovascular fitness between athletes. We did not collect details about the participants’ level of athleticism, such as sport, position, competition level, or individual-specific factors. All of these can affect how responsive an athlete’s cardiovascular ANS is and should be accounted for in future research.

## 6. Conclusions

In this retrospective chart review, we found that half of adolescents after SRC reported symptoms during postural change (OI) and that half reported the more-than-mild worsening of symptoms during low-intensity exercise (Early EI), yet there was no association between the two measures. Although both impairments are suspected to be due to post-concussion autonomic impairment, they may affect different aspects of cardiovascular autonomic regulation or illicit symptoms from non-ANS-related impairments, which warrants additional research. Nevertheless, OI identified on physical examination within the first week after SRC was significantly associated with delayed recovery, even without being associated with Early EI (another risk factor for delayed recovery). We believe this justifies the clinical utility of performing both tests at the initial assessment. The OI test is easy to integrate into a standard physical examination and has become a recommended part of the SCOAT for the best-practice management of SRC.

## Figures and Tables

**Figure 1 diagnostics-14-02743-f001:**
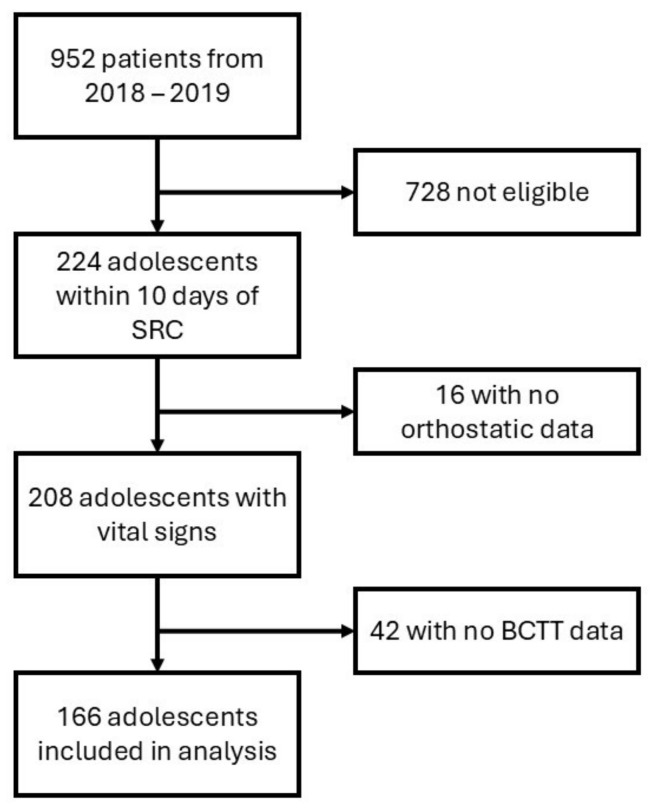
Sample inclusion flowchart.

**Table 1 diagnostics-14-02743-t001:** Sample demographics and clinical characteristics.

Demographics and Recovery Outcomes		
*n*		166
Age in years, mean (95% CI)		15.42 (15.19, 15.64)
Sex, *n* (%)	Female	69 (41.6%)
	Male	97 (58.4%)
History of Previous Concussion, *n* (%)	0	82 (49.4%)
	1	53 (31.9%)
	2	27 (16.3%)
	3+	4 (2.4%)
Injury to visit time in days, mean (95% CI)		5.55 (5.21, 5.89)
Total symptom severity ^1^ (max = 132)		32.43 (29.50, 35.36)
Recovery time in days, mean (95% CI)		19.93 (17.69, 22.18)
Delayed recovery (>28 days to recover), *n* (%)		27 (16.3%)
Clinical Exam Signs and Symptoms, mean (95% CI) or *n* (%)		
Nausea (max = 6)		0.98 (0.76, 1.21)
Balance (max = 6)		1.33 (1.13, 1.53)
Dizziness (max = 6)		1.65 (1.41, 1.89)
Total symptom severity ^1^ (max = 132)		32.43 (29.50, 35.36)
Abnormal VOR		89 (58.2%)
Abnormal CTG		89 (58.2%)
OH		14 (8.5%)
Orthostatics and Graded Exertion Testing, mean (95% CI)		
Supine Systolic BP in mmHg		119.35 (117.37, 121.32)
Supine Diastolic BP in mmHg		70.49 (69.44, 71.54)
Supine HR in bpm		66.23 (64.40, 68.06)
Standing Systolic BP in mmHg		117.92 (116.12, 119.72)
Standing Diastolic BP in mmHg		75.64 (74.47, 76.82)
Standing HR in bpm		77.82 (75.78, 79.86)
Mean Systolic Change in mmHg		−1.43 (−2.96, 0.10)
Mean Diastolic Change in mmHg		5.17 (4.02, 6.32)
Mean HR Change in bpm		11.59 (9.59, 13.59)
HRt on BCTT in bpm		134.67 (131.10, 138.24)
Exercise time on BCTT in minutes		8.09 (7.47, 8.71)
Exercise Tolerant, *n* (%)		24 (14.5%)
Main Outcome Measures, *n* (%)		
OI		80 (48.2%)
Early EI		87 (52.4%)

CI: confidence interval; BP: blood pressure; HR: heart rate; BCTT: buffalo concussion treadmill test; VOR: vestibulo-ocular reflex; CTG: complex tandem gait; OH: orthostatic hypotension; OI: orthostatic intolerance; EI: exercise intolerance; ^1^: symptom severity from Post-Concussion Symptom Inventory.

**Table 2 diagnostics-14-02743-t002:** Proportion of adolescents with and without OI and Early EI.

OI Screen	Early EI (HRt < 135 bpm) on BCTT		
	Positive	Negative	Total
Positive	47 (28.3%)	33 (19.9%)	80
Negative	40 (24.1%)	46 (27.7%)	86
Total	87	79	166

Data is presented as sample size (percent of total sample).

**Table 3 diagnostics-14-02743-t003:** Comparison of participants with and without OI.

		With OI	Without OI	*p*-Value
*n*		80	86	-
Age in years, mean (95% CI)		15.39 (15.08, 15.70)	15.44 (15.11, 15.77)	0.835
Sex, *n* (%)	Female	29 (37.2%)	34 (45.3%)	0.306
	Male	49 (62.8%)	41 (54.7%)	
History of Previous Concussion, *n* (%)	0	40 (50.0%)	42 (48.8%)	0.326
	1	24 (30.0%)	29 (33.7%)	
	2	12 (15.0%)	15 (17.4%)	
	3+	4 (5.0%)	0 (0.0%)	
Injury to visit time in days, mean (95% CI)		5.43 (4.94, 5.91)	5.67 (5.19, 6.16)	0.472
Recovery time in days, mean (95% CI)		23.59 (19.46, 27.73)	16.53 (14.73, 18.33)	**0.002**
Delayed recovery (took >28 days to recover), *n* (%)		19 (23.8%)	8 (9.3%)	**0.012**
Clinical Exam Signs and Symptoms, mean (95% CI) or *n* (%)				
Nausea (max = 6)		1.41 (1.04, 1.77)	0.59 (0.34, 0.83)	**<0.001**
Balance (max = 6)		1.57 (1.26, 1.88)	1.11 (0.86, 1.35)	**0.020**
Dizziness (max = 6)		2.04 (1.66, 2.41)	1.28 (0.99, 1.57)	**0.002**
Total symptom severity (max = 132)		37.53 (33.14, 41.92)	27.74 (24.01, 31.48)	**<0.001**
Abnormal VOR		54 (69.2%)	35 (46.7%)	**0.005**
Abnormal CTG		49 (62.8%)	40 (53.3%)	0.234
Orthostatics, mean (95% CI)				
Supine Systolic BP in mmHg		120.20 (117.08, 123.33)	118.56 (116.04, 121.08)	0.413
Supine Diastolic BP in mmHg		70.49 (69.08, 71.91)	70.49 (68.93, 72.05)	0.996
Supine HR in bpm		64.45 (61.84, 67.04)	67.88 (65.31, 70.45)	0.064
Standing Systolic BP in mmHg		117.73 (114.94, 120.51)	118.10 (115.74, 120.47)	0.836
Standing Diastolic BP in mmHg		75.45 (74.61, 80.82)	75.83 (74.28, 77.37)	0.753
Standing HR in bpm		77.71 (74.61, 80.82)	77.92 (75.17, 80.67)	0.921
Mean Systolic Change in mmHg		−2.49 (−4.58, −0.41)	−0.45 (−2.69, 1.79)	0.189
Mean Diastolic Change in mmHg		4.99 (3.07, 6.91)	5.34 (3.97, 6.70)	0.765
Mean HR Change in bpm		13.26 (10.82, 15.71)	10.03 (6.90, 13.17)	0.112
Graded Exertion Testing, mean (95% CI)				
HRt on BCTT in bpm		131.86 (126.56, 137.17)	137.30 (132.46, 142.15)	0.133
Exercise time on BCTT in minutes		8.01 (7.09, 8.94)	8.16 (7.33, 9.00)	0.811

Bolded values indicate a statistically significant difference at *p* < 0.05.

## Data Availability

Deidentified data is available from https://doi.org/10.7910/DVN/M7QSNL (accessed on 29 October 2024).

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
