# Peer review of "The Clinical Utility of Assessing Orthostatic Intolerance in Adolescents with Sport-Related Concussion, a Retrospective Study"

_diagnostics, 2024, doi:10.3390/diagnostics14232743_

Round 1

Reviewer 1 Report

Comments and Suggestions for Authors

Overall, I really like this manuscript and feel that it brings value to the overall field and to sport concussion-related clinical care. There remains a need for concussion markers for both diagnosis and recovery, with orthostatic testing and supervised exercise both areas where the optimal use has not yet been determined. Although this manuscript presents a negative finding, it is an important step nonetheless in assessing for relationships between findings from orthostatic and exercise testing. As noted, orthostatic testing has the advantage of being more easily utilized in any clinical setting, whereas most clinics seeing concussed patients will not have easy access to exercise testing, so an improved understanding of the diagnostic and prognostic utility of orthostatic testing is important.

I do not find it surprising that the present study found no significant association between positive orthostatic intolerance and exercise intolerance as defined, given the heterogeneic contributions to either finding. While I agree that both tests may have potential to show post-concussion autonomic dysfunction, I do think it is importantly stated that there may be other causes of symptoms with orthostatic testing, such as vestibular dysfunction as stated in lines 263-266. This is directly supported by the study as there is a significantly higher percentage of patients with OI who have abnormal VOR compared to those without OI. Therefore, my widest suggestion for improving this manuscript is to soften the strong language that more definitively suggests that OI and EI are due to autonomic dysfunction. In addition to vestibular dysfunction, I could surmise that there may be additional considerations that could explain symptom provocation with postural changes as well as exercise, such as cervicogenic factors or general concussion-related symptom burden (eg migrainous headache). I additionally recommend expanding the discussion of other possible explanations for positive testing.

Examples where I feel that the language connecting OI and/or EI soley to autonomic function can be softened:

-Lines 54-56

-Lines 66-67

-Lines 93-94

-The paragraph including lines 255-268 can be strengthened by opening up this portion of the discussion to include other possible contributors to OI besides autonomic dysfunction. 

Author Response

Comment 1: Overall, I really like this manuscript and feel that it brings value to the overall field and to sport concussion-related clinical care. There remains a need for concussion markers for both diagnosis and recovery, with orthostatic testing and supervised exercise both areas where the optimal use has not yet been determined. Although this manuscript presents a negative finding, it is an important step nonetheless in assessing for relationships between findings from orthostatic and exercise testing. As noted, orthostatic testing has the advantage of being more easily utilized in any clinical setting, whereas most clinics seeing concussed patients will not have easy access to exercise testing, so an improved understanding of the diagnostic and prognostic utility of orthostatic testing is important. I do not find it surprising that the present study found no significant association between positive orthostatic intolerance and exercise intolerance as defined, given the heterogeneic contributions to either finding. While I agree that both tests may have potential to show post-concussion autonomic dysfunction, I do think it is importantly stated that there may be other causes of symptoms with orthostatic testing, such as vestibular dysfunction as stated in lines 263-266. This is directly supported by the study as there is a significantly higher percentage of patients with OI who have abnormal VOR compared to those without OI. Therefore, my widest suggestion for improving this manuscript is to soften the strong language that more definitively suggests that OI and EI are due to autonomic dysfunction. In addition to vestibular dysfunction, I could surmise that there may be additional considerations that could explain symptom provocation with postural changes as well as exercise, such as cervicogenic factors or general concussion-related symptom burden (eg migrainous headache). I additionally recommend expanding the discussion of other possible explanations for positive testing. Examples where I feel that the language connecting OI and/or EI soley to autonomic function can be softened:-Lines 54-56-Lines 66-67-Lines 93-94

Response 1: Thank you for reviewing our manuscript and providing us with your suggestions. We have incorporated all of them into the manuscript. We have softened the language in those sentences and added an additional paragraph about other causes for symptom exacerbations on orthostatics and on exertion testing in the discussion. Please see lines 54, 56-58, 69-70, 96 and 268-278.

Comment 2: -The paragraph including lines 255-268 can be strengthened by opening up this portion of the discussion to include other possible contributors to OI besides autonomic dysfunction. 

Response 2: Addressed in the previous comment. Thank you again.

Reviewer 2 Report

Comments and Suggestions for Authors

The authors have conducted an interesting study. The assessment includes an assessment of the significance of orthostatic intolerance testing in adolescents with concussion.

The work is interesting and well-conducted. I have a few comments, the introduction of which may significantly improve the quality of the work:
- There is a lack of a more complete characterization of the population in terms of physical activity. It is known that aging athlete's heart presents different echocardiographic features of competitive sprint- vs endurance-trained master athletes. I believe that these mechanisms should be outlined and appropriately related to the type of physical activity.
- whether changes were observed in the diagnostics conducted, whether personal monitoring measures were used, e.g. currently popular smart watches.

I believe that a slight modification of the work will allow for its publication to be considered.

Author Response

Comment 1: The authors have conducted an interesting study. The assessment includes an assessment of the significance of orthostatic intolerance testing in adolescents with concussion.

Response 1: Thank you for reviewing our manuscript. Our changes are marked in red in the revised manuscript. We need clarification for the comment below, we will make the changes in the second round.

Comment 2: The work is interesting and well-conducted. I have a few comments, the introduction of which may significantly improve the quality of the work:
- There is a lack of a more complete characterization of the population in terms of physical activity. It is known that aging athlete's heart presents different echocardiographic features of competitive sprint- vs endurance-trained master athletes. I believe that these mechanisms should be outlined and appropriately related to the type of physical activity.

Response 2: Sorry, but we need a little more clarification about where we should add this. Our patient population consisted of school-age adolescents who were injured in organized or recreational sports (mentioned in lines 110-112). If you are asking about their primary sport or sport of injury, then unfortunately we cannot provide this since it is not consistently reported on our physician's progress notes. We can add this as a limitation is necessary. 

Comment 3: - whether changes were observed in the diagnostics conducted, whether personal monitoring measures were used, e.g. currently popular smart watches.
I believe that a slight modification of the work will allow for its publication to be considered.

Response 3: We have added details about the devices used at our clinic, please see lines 131, 143-145. All blood pressure assessments were performed using a Welch Allyn automated cuff, the treadmill was a Landice L8 and the HR monitor was the Polar OH1 (which is now called the Verity sense). Thank you again for reviewing our manuscript.

Round 2

Reviewer 2 Report

Comments and Suggestions for Authors

The authors have made corrections to the manuscript. I understand that if some data were not collected, they cannot be included :-) however, I would include such information in the study limitations. In my opinion, after adding this information, the manuscript can be considered for publication.

Author Response

Thank you, we have added it as a limitation. Please see line 315-320.